# Migrasome-Related Genes as Potential Prognosis and Immunotherapy Response Predictors for Colorectal Cancer

**DOI:** 10.3390/biomedicines13040799

**Published:** 2025-03-26

**Authors:** Lu Chang, Chao Qin, Yimin Chu, Ming Guan, Xuan Deng

**Affiliations:** 1Department of Laboratory Medicine, Huashan Hospital Fudan University, Shanghai 200040, China; 17865327413@163.com (L.C.); qinchaoemail@163.com (C.Q.); 2Digestive Endoscopy Center, Tongren Hospital, Shanghai Jiaotong University School of Medicine, Shanghai 200050, China; yimin_chu@163.com

**Keywords:** colorectal cancer, migrasome, prognosis, tumor immunity, scRNA-seq, T1MP1, CXCL8, MGP

## Abstract

**Background:** Studies highlight the role of migrasomes as mediators of intercellular communication and signaling, critical in influencing tumorigenesis and progression. Yet migrasome-related genes and their potential role in colorectal cancer prognosis remain unexplored. **Methods:** Differentially expressed gene set A (DEG set A) was identified in the TCGA-CRC dataset, and Weighted Gene Co-expression Network Analysis (WGCNA) was performed to identify the most important modules associated with migrasome-related gene (MRG) scores. Single-cell RNA-seq dataset GSE231559 DEG set B was determined. Candidate migrasome-related genes were filtered by intersecting DGE set A, key module genes, and DEG set B. Prognostic genes were subsequently screened through regression analysis, and a risk model was developed. Patients with CRC in the TCGA cohort were stratified into high- and low-risk groups based on the optimal cutoff of the risk score. Immunotherapy response-related analyses were then performed. Finally, cell-to-cell communication analysis was carried out for key cells identified based on prognostic gene expression analysis in annotated cells. **Results:** The six candidate migrasome-related genes were identified through the overlap of 5158 DEG set A, 1960 key module genes, and 146 DEG set B. Further screening led to the selection of T1MP1, CXCL8, and MGP as potential prognostic biomarkers. Immune-related analysis indicated that the high-risk group exhibited a better response to immunotherapy. Notably, the prognostic genes showed elevated expression levels in monocytes and tissue stem cells, thereby designating them as key cell types. **Conclusions:** We conducted bioinformatic analysis of migrasome-related genes and identified significant involvement of T1MP1, CXCL8, and MGP in influencing CRC prognosis and immunotherapy response. Our research provides novel insights into the role of migrasomes in CRC biology.

## 1. Introduction

Colorectal cancer (CRC) is the third most frequently diagnosed cancer worldwide and the second major cause of cancer-related mortality, representing roughly 10% of all cancer cases and fatalities [1]. Around 70% of patients diagnosed with CRC exhibit AJCC Stage I–III. For these patients, surgical resection combining chemotherapy remains the primary treatment with curative aim, showing a 5-year survival rate ranging from 91% for those with localized disease to 73% for those with regional disease [2]. Patients with CRC diagnosed at advanced stages or relapse after tumor resection generally have a more unfavorable prognosis; the 5-year survival rate drops drastically to only 14% [3]. Immunotherapy has been extensively developed by harnessing the patient’s immune system to target and combat cancer cells, significantly revolutionizing the way advanced-stage cancers are treated [4,5]. Unfortunately, only a small proportion of patients with CRC respond to immunotherapy [6,7]. Thus, identifying patients at elevated risk of relapse post-treatment and most prone to exhibit favorable responses to immunotherapy could improve clinical management and facilitate the customization of therapy options.

Migrasomes are membrane-bound organelles released from the cell membrane as a result of the mechanical forces generated during cell migration [8]. Migrasomes function as a means for cells to eliminate unwanted cellular material, consisting of lipid-rich vesicles that enclose various cellular components, including proteins, RNAs, and lipids [9]. They are crucial in cellular processes, acting as essential mediators for the exchange of signals between tumor cells and the tumor microenvironment [10]. Migrasomes play a key role in multiple facets of tumor-related processes like immune system regulation, vascular permeability, and extracellular matrix remodeling [11]. During the process of tumor metastasis, cancer cells dynamically interact with vascular endothelial cells as they migrate through the circulatory system. This interaction may facilitate the extensive production of migrasomes. Once released into the circulation, these migrasomes could play a pivotal role in reshaping the tumor immune microenvironment by modulating the functions of immune cells [12]. Research has revealed that pancreatic cancer cells are capable of generating migrasomes, which can be taken up by macrophages. This uptake promotes the formation of an immunosuppressive tumor microenvironment, thereby accelerating tumor progression and metastasis [13]. Studies have shown that the migrasome regulator TSPAN4 can directly enhance tumor cell proliferation and induce the polarization of tumor-associated macrophages toward the immunosuppressive M2 phenotype, thereby establishing an immunosuppressive microenvironment [14], which in turn promotes tumor progression. Additionally, TSPAN8, as a critical angiogenesis inducer, significantly inhibits the tube-forming ability of HUVECs in vitro when its expression is knocked down in tumor cells [15]. Despite their significant implications in tumor biology, research on migrasomes in colorectal cancer (CRC) remains limited. Studies focusing on migrasome-related genes in CRC are scarce, highlighting a need for further investigation in this area.

This study utilizes data from the TCGA and GEO cohorts for CRC. Using bioinformatics approaches, we identified and validated prognostic genes associated with migrasomes in CRC. Additional comprehensive analyses, such as single-cell analysis, functional enrichment, survival analysis, and immune infiltration, were performed to explore the potential mechanisms driving CRC. Our research establishes a robust theoretical framework for the function of migrasomes in CRC biology.

## 2. Materials and Methods

### 2.1. Data Collection

The Cancer Genome Atlas-Colorectal Cancer (TCGA-CRC) dataset, which comprises 644 CRC samples and 44 normal tissue samples, provided the transcriptome information and survival information of patients with colorectal cancer (CRC). This dataset was used as the test cohort and was accessible via the TCGA database (https://www.cancer.gov/ccg/research/genome-sequencing/tcga, accessed on 5 June 2023). Furthermore, for validation purposes, the GSE17537 dataset—which includes 55 CRC tissue samples (including 50 individuals of European ancestry, 4 individuals of African ancestry, and 1 individual of Hispanic/Latino ethnicity) and is based on the GPL570 platform—was obtained from the Gene Expression Omnibus (GEO) database (http://www.ncbi.nlm.nih.gov/geo/, accessed on 15 March 2023). Additionally, the GSE231559 dataset was used to extract single-cell RNA sequencing (scRNA-seq) data, which includes 6 CRC and 3 control tissue samples, and was extracted from the GPL18573 platform in the GEO database. Seven migrasome-related genes (MRGs) were identified from the literature [12,16].

### 2.2. Differential Expression Analysis

The TCGA-CRC dataset’s differentially expressed genes (DEGs) in set A (CRC vs. control) were found using the “DESeq2” program (v1.34.0) [17], with the screening criterion |log2FoldChange (FC)| > 0.5 and *p* < 0.05. The “ggplot2” package (v3.4.1) [18] was used to create volcano plots for DEG set A, and the “ComplexHeatmap” package (v2.14.0) [19] was used to create heatmaps of the top 10 upregulated and top 10 downregulated genes in set A.

### 2.3. Identified the Migrasome-Related Score

Using the “GSVA” software (v 1.46.0), the single-sample gene set enrichment analysis (ssGSEA) technique was used to calculate the MRG scores for each sample in the TCGA-CRC dataset based on seven migrasome-related genes (MRGs) [20]. Subsequently, an analysis was performed on the difference in MRG scores between the control and CRC samples (*p* < 0.05). The median MRG score was used to stratify patients with CRC into high- and low-scoring groups. To evaluate survival differences between these groups, Kaplan–Meier (KM) analysis was performed; the log-rank test (*p* < 0.05) was used to define statistical significance.

In the TCGA-CRC dataset, the single-sample gene set enrichment analysis (ssGSEA) algorithm was used to calculate the MRG scores for all samples based on 7 migrasome-related genes (MRGs) using the “GSVA” package (v 1.46.0) [20]. Subsequently, the difference in MRG scores between CRC and control samples was analyzed (*p* < 0.05). Based on the median MRG score, patients with CRC were classified into high- and low-score groups. Kaplan–Meier (KM) analysis was then performed to evaluate the survival differences between the two groups, with the *p* value determined using the log-rank test (*p* < 0.05).

### 2.4. Identification of Gene Modules via Weighted Gene Co-Expression Network Analysis (WGCNA)

Using the “WGCNA” R package (v 1.71), WGCNA was applied to identify the key module most closely linked to MRG scores in the TCGA-CRC dataset [21]. First, all samples from TCGA-CRC were clustered to remove outliers. To ensure that the network had scale-free properties while avoiding low connectivity, R^2^ = 0.85 was set as the target, and a soft threshold β that satisfied the R^2^ requirement and had connectivity close to 0 was chosen as the optimal value. Then, a co-expression matrix with a minimum module size of 50 genes was constructed. Highly similar modules were merged based on a phase specificity threshold of 0.2, and distinct colors were assigned to different gene modules. The correlation coefficient between MRG scores and gene modules was calculated, and the module with the strongest association was selected. Spearman’s correlation was then used to assess the relationship between gene significance for MRG scores and module membership within the key module, from which candidate genes were identified.

### 2.5. Examination of Data from Single-Cell RNA Sequencing (scRNA-Seq)

Samples from the dataset GSE231559 were integrated for scRNA-seq analysis employing the “Seurat” package (v 5.2.1) [22]. Initially, low-quality cells were removed following quality control (QC) standards, with cells being filtered using thresholds of nCount-RNA < 20,000, and percent.mt < 5%, 200 ≤ nFeature-RNA ≤ 7000. The “FindVariableFeatures” function was used to identify the 2000 most variable genes (HVGs) following data normalization using the “NormalizeData” function. These 2000 HVGs were then subjected to principal component analysis (PCA), and the principal components (PCs) were determined. Next, the cells were clustered using Projection (UMAP) and Uniform Manifold Approximation. Following this, cells were annotated based on marker genes identified through literature mining [23] using the “singleR” package (v 1.831) [24]. Additionally, the proportion of annotated cells in the CRC tumor and control samples was visualized separately.

### 2.6. Function Analysis and Differential Expression Analysis

The cells were divided into high- and low-scoring groups according to the median score in the GSE231559 dataset, and the MRG scores were computed using ssGSEA. Subsequently, DEG set B (high-scoring vs. low-scoring) was detected using the “DESeq2” package (v 1.47.5), applying thresholds of |log2FC| > 0.5 and adj. *p*. Val < 0.05. Volcano plots and heatmaps were subsequently created to display the results. Then, using the “VennDiagram” software (v 1.7.3), candidate genes were found by overlapping DEG set A, important module genes, and DEG set B [25]. Using the “clusterProfiler” package (v 4.2.2), studies of the Kyoto Encyclopedia of Genes and Genomes (KEGG) and Gene Ontology (GO) were performed to investigate the biological processes and signaling pathways associated with the candidate genes [26], using *p* < 0.05 as the significance level. Furthermore, using the STRING database (https://string-db.org/, accessed on 15 March 2023), a protein-protein interaction (PPI) network was constructed with a confidence score of 0.4 in order to examine the protein interactions of the candidate genes. Cytoscape (v 3.9.1) was used to show the PPI network after isolated proteins were eliminated [27].

### 2.7. Creation and Approval of the Risk Model

To find predictive markers in TCGA-CRC, univariate Cox analyses (*p* < 0.05) and Least Absolute Shrinkage and Selection Operator (LASSO) regression (family = Cox) were performed using the “survival” (v 3.3-1) [28] and “glmnet” (v 4.1-6) [29] packages based on the candidate genes. A risk model was then created using the prognostic genes that were discovered. Using the following formula, the risk score was determined:$$riskscore=\sum_{i=1}^{n}\left({coef}_{i}*x_{i}\right)$$
whereas “x” stands for the expression levels of the prognostic genes, and “coef” indicates their coefficients. Patients with CRC in the TCGA-CRC and GSE175371 datasets were divided into high-risk and low-risk groups based on the ideal threshold of the risk score in order to assess and verify the risk model. Risk curves for the two cohorts were then plotted in both datasets. Moreover, Kaplan–Meier (K-M) curves were created using the “survminer” package (v 0.4.9) to assess the overall survival (OS) of patients with CRC in both datasets (*p* < 0.05) [30]. Concurrently, the “survivalROC” program (v 1.0.3) was used to create receiver operating characteristic (ROC) curves for 1- to 3-year survival; in both datasets, the Area Under the Curve (AUC) values were more than 0.6 [31].

### 2.8. Nomogram Was Established and Validated

Clinical characteristics, including age and gender, were common clinical features that might influence the incidence and prognosis of colorectal cancer [32]; TNM staging was an important tool for assessing tumor size and extent of spread and was crucial for predicting survival in patients with CRC [33]. Using the TCGA-CRC dataset, the risk score and clinical covariates (age, gender, N, T, and M stages) were included in the proportional hazards (PH) assumption test (*p* > 0.05), multivariate Cox analysis (*p* < 0.05), and univariate Cox analysis (*p* < 0.05). A nomogram based on the independent prognostic factors was created using the “rms” package (v 6.2-0) to predict 1-, 2-, and 3-year survival in patients with colorectal cancer. Independent prognostic factors were then identified, and the “rms” package (v 6.2-0) was used to create a nomogram that predicted the 1-, 2-, and 3-year survival of patients with colorectal cancer [30]. The accuracy of the nomogram was then assessed using the calibration curve, decision curve analysis (DCA), and ROC curves. Additionally, the “survival” program was used to analyze the variations in survival between the high-risk and low-risk groups in order to ascertain how relevant the risk score was to various clinical factors based on clinical features in the TCGA-CRC dataset. These clinical characteristics included gender (male and female), age (>60 and ≤60), N stage (N0 and N+), and T phases (T0–T1 and T3–T4).

### 2.9. Analysis of Gene Set Enrichment (GSEA)

Functional enrichment analysis was conducted to further explore the prognostic genes. Spearman correlation coefficients were calculated between each prognostic gene and all other genes, then ranked from largest to smallest. Additionally, the background gene set used was “c2.cp.v7.2.symbols.gmt” from the Molecular Signatures Database (MSigDB, https://www.gsea-msigdb.org/gsea/msigdb/index.jsp, accessed on 10 June 2023). Following this, the screening cutoff for Analysis of Gene Set Enrichment (GSEA) was adj. *p* < 0.05. The top 6 enriched pathways were visualized using the “enrichplot” package (v 1.18.0) [34].

### 2.10. Analysis of Immunological Checkpoints and Immune Infiltration

For patients with CRC in both high- and low-risk groups, the stromal score, immunological score, and ESTIMATE score were determined using the ESTIMATE algorithm (v 1.0.13) [35] in the TCGA-CRC dataset. Furthermore, using ssGSEA, the infiltration levels of 28 immune cell types in these cohorts were determined. Immune infiltration scores were compared between the high-risk and low-risk groups using the Wilcoxon test (*p* < 0.05). The association between immune cell population changes and prognostic genes was assessed using Spearman correlation analysis (|r| > 0.50 and *p* < 0.05).

### 2.11. Regulation Network and Subcellular Localization Analyses

Genes involved in immunogenic cell death and those encoding human leukocyte antigens (HLAs) are essential for the immune processes of tumors. In this study, 27 immunogenic cell death genes (CD276, TNFRSF8, CD44, etc.) and 7 human leukocyte antigen genes (including HLA-G, HLA-DRB1, HLA-DQA1, HLA-DRA, HLA-DMB, HLA-DPA1, HLA-DMA and HLA-DPB1, and HLA-DOA) were employed to examine the variations in expression between the high- and low-risk categories in the TCGA-CRC dataset.

Furthermore, immunological checkpoint inhibitors have significant therapeutic uses in immunotherapy and are essential for regulating immune function. These eight common immune checkpoint genes, PDCD1-LG2 (PD-1LG2), LGALS9 (GAL9), PDCD1 (PD-1), CD274 (PD-L1), HAVCR2 (TIM-3), LAG-3, TIGHT, and CTLA-4, were chosen for this investigation. Immunological checkpoints were assessed for differential expression between high- and low-risk groups (*p* < 0.05). With a correlation criterion of |r| > 0.30 and *p* < 0.05, Spearman correlation analysis was then employed to ascertain the relationships between the differential immunological checkpoints. In addition, based on the median score of differential immune checkpoints, the sample of patients with CRC was divided into high- and low-expression groups, and KM survival analysis was performed using the “survminer” package (v 0.4.9), and the survival curves of the two groups were compared using the log-rank test (*p* < 0.05) [30].

### 2.12. Analysis of Gene Expression and Drug Sensitivity

The “pRRophetic” program (v 0.5) was used to analyze drug sensitivity among people with CRC at high- and low risk based on the Genomics of Drug Sensitivity in Cancer (GDSC) database (https://www.cancerrxgene.org/, accessed on 15 August 2023) [36] Using a cutoff of |r| > 0.6 and *p* < 0.05, the association between prognostic genes and medication sensitivity was examined.

Additionally, predictive gene expression was examined in TCGA-CRC control samples and CRC samples (*p* < 0.05). Further research was carried out on the protein-level expression of these prognostic genes in CRC and control samples from TCGA-CRC using the Human Protein Atlas (HPA) database (http://www.alzdata.org/, accessed on 5 September 2023).

### 2.13. Identification of the Key Cells

Prognostic gene expression in CRC samples was visualized using UMAP in comparison to control samples across annotated cells. The identification of key cells was predicated on the presence of high expression levels and differential expression in at least two prognostic genes. To investigate the biological pathways and activities that these important cells could be connected to, enrichment analysis was conducted using the “ReactomeGSA” program [37].

### 2.14. Cell-to-Cell Communication and Pseudo-Temporal Analyses

To comprehend the important cells more thoroughly, the cells’ cell-to-cell communication was analyzed using the “CellChat” program (v 1.6.1) [38]. Additionally, to analyze the trajectories of the significant cells and examine the differentiation stages, the “Monocle2” software (v 2.26.0) was used to perform pseudo-temporal analysis [39]. The dynamic trends of prognostic gene expression during key cell differentiation were visualized using the “plot genes in pseudotime” function.

### 2.15. Statistical Analysis

For all statistical analyses in this paper, R (v 4.1.0) was used. Unless otherwise noted, a *p*-value of less than 0.05 was regarded as statistically significant.

## 3. Results

### 3.1. Identification of Migrasome-Related Module Genes

For the TCGA-CRC dataset, a total of 5158 DEGs (CRC vs. control) (labeled as DEG set A) were screened. Among these DEGs, 2623 were upregulated and 2535 were down-expressed (Figure 1a,b). MRG scores were calculated based on the expression of seven migrasome genes (ITGB1, ITGA5, EOGT, CPQ, PIGK, NDST1, and TSPAN4) [12]. We separated CRC samples into groups with high MRG score and low score; the Kaplan–Meier curve showed that the group with low MRG score demonstrated a markedly increased overall survival (OS) probability (Figure 1c,d). Then, a co-expression matrix of MRG was established using the R package WGCNA. No outlier samples were identified (Appendix A). In our work, β = 26 (R^2^ of the scale-free network = 0.85) was screened as the soft thresholding parameter (Appendix A). As demonstrated in Figure 1e and Appendix A, eight co-expressed gene modules were determined. The red module had the strongest connection with MRGs score (r = 0.85 and *p* < 0.05), containing 960 key module genes (Figure 1f). Thus, using ssGSEA and WGCNA analysis, 7 migrasome-related genes (MRGs) were expanded to 960 migrasome-related module genes (WGCNA-MRGs).

### 3.2. Functional Enrichment Study of the Six Migrasome-Related Genes

For the GSE231559 dataset, ineligible cells were filtered, yielding 1887 core cells and 24,193 genes for subsequent analysis. The top 10 highly variable genes were presented (Appendix A). Following this, PCA was performed, the samples were well integrated in the first two PCs (Appendix A), and the top 40 PCs were selected for subsequent analysis (Appendix A). Using UMAP analysis, all cells were classified into distinct clusters and were annotated (Figure 2a). Different cell types displayed variant proportions between normal and tumor tissues, with B cells exhibiting a higher ratio while T cells had a lower ratio in CRCs (Figure 2b,c).

Then the 1887 cells were divided into groups: the high MRG scoring (943 cells) and low MRG scoring (944 cells) groups, according to the expression of migrasome genes (Appendix A). In total, 146 DEGs (high-scoring vs. low-scoring) (labeled as DEG set B) were identified, comprising 78 upregulation and 68 downregulation genes (Figure 2d). We overlapped DEG set A containing 5158 genes, 1960 key module genes, and DEG set B containing 146 genes; 6 candidate genes were identified, including TIMP1, CXCL8, MGP, VCAN, ZEB2, and IL1B (Figure 2e).

Function enrichment analysis of the above-mentioned 6 candidate genes was performed; 286 GO terms and 20 KEGG pathways were selected and determined by the threshold of *p* < 0.05. Specifically, GO terms indicated that intersection genes were mainly associated with cell adhesion molecule production, collagen-containing extracellular matrix, cytokine activity, etc. (Figure 2f). The 20 KEGG pathways comprised IL-17, NF-kappa B, and Toll-like receptor signaling pathway (Figure 2g). In addition, PPI networks provide interaction relationships detected between pairs of candidate proteins (Figure 2h).

### 3.3. Prognostic Performance Analysis of the Risk Model

Subsequently, univariate Cox analysis was conducted based on the correlation between levels of six candidate genes and OS in the TCGA-CRC cohort. We showed that expression of CXCL8 (*p* = 0.036), MGP (*p* = 0.016), and TIMP1 (*p* < 0.001) was significantly correlated with OS (Figure 3a). CXCL8, MGP, and TIMP1 were further selected as prognostic model genes by LASSO analysis (lambda_min_ = 0.039). Thus, we developed a prognostic risk model using these 3 prognostic genes (Appendix A). We scored patients with CRC according to the risk coefficient obtained by LASSO regression analysis and stratified them into high- and low-risk groups by the median score (Appendix A). OS survival curve was drawn and showed that high-risk patients exhibited shorter OS time (*p* < 0.0001) (Figure 3b). The Area Under Curve (AUC) values of ROC are all above 0.6, suggesting that the risk model demonstrates satisfactory performance (Figure 3c). The effectiveness of the risk model was further validated using the GSE17537 dataset (Appendix A).

In the TCGA-CRC dataset, univariate Cox analysis identified prognostic factors including RiskScore (based on expression of TIMP1, MGP, and CXCL8), age, N stage, and M stage (Figure 4a). The independent prognostic factors (age, N stage, and M stage) were further determined by multivariate Cox analysis (*p* < 0.05) and the proportional hazards (PH) assumption test (*p* > 0.05) (Figure 4b). A nomogram was then constructed based on independent prognostic factors (age, N-stage, and M-stage), with higher points implying lower survival in patients with CRC (Figure 4c). The slope of the calibration curve (1, 2, and 3 years) for the nomogram was close to 1, indicating that the nomogram was a good predictor (Appendix A). Additionally, the decision curve analysis (DCA) values indicated that the net income of the nomogram surpasses that of the single factor, suggesting a superior predictive effect of the nomogram (Appendix A). The AUC values for 1, 2, and 3 years were then calculated and shown to exceed 0.7, demonstrating that the nomogram was a reliable predictor for CRC prognosis (Figure 4d).

To evaluate whether the risk score is applicable to different clinicopathological features, an analysis of stratified survival between high- and low-risk groups across diverse clinicopathological features was conducted. As shown in Figure 5a, survival disparities were noted in the subgroups of individuals over 60 years and those 60 years or younger. There was no notable survival disparity in the male, pathological N+ or T0–T1 grouping, whereas a significant survival difference was observed in the female, pathological N0 and T2–T3 subgroup.

To investigate the underlying biological mechanisms of prognostic genes, GSEA was conducted on the TCGA-CRC. T1MP1 and MGP were found to be co-enriched in pathways related to “aminoacyl trna biosynthesis”, “systemic lupus erythematosus” (Figure 5b,c). Furthermore, CXCL8 was associated with “ribosome”, “graft versus host disease”, and “butanoate metabolism”, etc. (Figure 5d).

### 3.4. Immunotherapy Response Prediction of the Risk Model

It is reported that contents released from migrasomes during breakdown have a significant impact on the local immune modulation of cancers. We therefore delineated the immune cell infiltration landscape inside the TME of CRC across different groups (based on expression of TIMP1, MGP, and CXCL8 risk genes). Patients with CRC with high risk exhibited higher stromal score and ESTIMATE score, while no significant difference was observed in immune scores between the high- and low-risk cohorts (Appendix A). The ratio of 28 immune cells in high- and low-risk cohorts is presented in Appendix A. Then, 13 differential immune cells were selected with *p* < 0.05, including activated natural killer (NK) cells, neutrophils, and CD8^+^ T cells (Figure 6a). In addition, prognostic genes exhibited a substantial correlation with several distinct immune cells. Specifically, NK cells showed significant correlation with MGP (r = 0.69, *p* < 0.01), activated CD4 T cells with CXCL8 (r = 0.54, *p* < 0.01), and NK T cells with TIMP1 (r = 0.52, *p* < 0.01), among others (Figure 6b).

Furthermore, 16 immunogenic cell death genes and 5 human leukocyte antigen genes were separately identified as their expression was correlated with the risk group of patients with CRC (Figure 7a,b). Furthermore, five immune checkpoints were found with significantly differential expression between the two risk cohorts (Figure 7c). Correlation analysis demonstrated strong connection between PDCD1 and LAG3 (r = 0.81), PDCD1 and CD274 (r = 0.59), and CD274 and LAG3 (r = 0.68) (Figure 7d). Further analyses showed that the high- and low-expression profiles of CD274, LAG3, LGALS9, and PDCD1 were significantly correlated with the prognostic survival of the patient samples with CRC, while the high and low expression of TIMP3 was not significantly correlated with the prognostic survival (Appendix A). Moreover, a better immunotherapy response rate was observed in the cohort with high risk, and the Tumor Immune Dysfunction and Exclusion (TIDE) value of CRC samples with high risk was higher compared to the cohort with low risk (Figure 7e). Additionally, the AUC value exceeded 0.6, signifying that the prediction for immunotherapy response by our risk model was effective (Figure 7f). The predictive efficacy of the risk model for immunotherapy response was further validated by an AUC value exceeding 0.6, indicating that the risk score prediction for immunotherapy response in the TCGA-CRC dataset was effective (Figure 7f).

### 3.5. Drug Sensitivity Analysis and Expression Validation

Through correlation analysis of prognostic genes and drug sensitivity, we identified a strong association between MGP and AP.24534 (Figure 8a). Furthermore, expression analysis of prognostic genes in TCGA-CRC revealed that CXCL8 and TIMP1 were significantly upregulated in CRC samples, whereas MGP showed markedly reduced expression (Figure 8b). Consistent with these findings, we observed that CXCL8 and TIMP1 protein levels were also significantly elevated in CRC samples (Figure 8c).

### 3.6. Analysis of 3 Migrasome-Related Prognostic Genes at the Single-Cell Level

We then analyzed the expression preference of three prognostic genes in the GSE231559 single-cell RNA-seq dataset. UMAP plots showed that TIMP1 and CXCL8 exhibited the most drastic monocyte and tissue stem cell preference; MGP was found to have a high expression in tissue stem cells (Figure 2a and Figure 9a,b). Therefore, monocytes and tissue stem cells were identified as key cells for the three prognostic genes. Signature gene enrichment analysis of key cells showed that monocyte was enriched in “Proton/oligopeptide cotransporters”, “Cobalamin (Cbl) metabolism”, and “ABO blood group systems biosynthesis”. Tissue stem cell was significantly related to “intracellular oxygen transport”, “ATP-sensitive Potassium channels”, and “Histamine receptors” (Figure 9c).

Communication between monocytes and tissue stem cells was observed by cell–cell communication analysis (Figure 9d). Furthermore, pseudo-temporal analysis was performed, monocyte was divided into 5 states (state 1–5) and 3 branches, and tissue stem cell was divided into 5 states (state 1–5) and 5 branches (Figure 9e). The expression levels of CXCL8 and TIMP1 during monocyte differentiation initially increased and then declined (Figure 9f). Levels of CXCL8, MGP, and TIMP1 during tissue stem cell differentiation initially increased, then declined, and subsequently rose again (Figure 9g).

## 4. Discussion

Colorectal cancer (CRC) is characterized by complex interactions between tumor cells and the tumor microenvironment (TME). Migrasomes, emerging vesicular structures, play a critical role in mediating tumor–stroma cross-talk through selective cargo transfer, such as oncogenic miRNAs (e.g., miR-21, miR-155) [40] and inflammatory cytokines (e.g., TGF-β, IL-8) [41], which promote immune evasion, angiogenesis, and metastatic niche formation in various cancers. Our pathway analysis revealed that migrasome-related genes are significantly associated with processes like “extracellular vesicle biogenesis” and “chemokine signaling”, aligning with their proposed functions in TME remodeling. Despite growing insights into migrasomes in various cancers, their role in CRC remains underexplored. This study aims to elucidate the functional significance of migrasomes in CRC and evaluate their potential as biomarkers or therapeutic targets.

Although tetraspanins play a crucial role in migrasome formation and function across various cancers, the specific tetraspanins involved can vary significantly depending on the cancer type. For instance, in hepatocellular carcinoma (HCC), TSPAN9 appears to be more prominently associated with migrasome activity [42], whereas in colorectal cancer (CRC), CD151 emerges as a central player [43]. This divergence in tetraspanin dependency highlights the tissue-specific and cancer-type-specific mechanisms underlying migrasome biology. Here, we aimed to identify migrasome-related genes and pathways and evaluate their relevance to CRC prognosis. Among the identified prognostic genes, TIMP1, CXCL8, and MGP have been reported in various cancers. Known for its role in aggressive cancers, TIMP1 promotes macrophage polarization via IL-10 and TGF-β secretion, enhancing tumor progression through the PI3K/AKT pathway [44]. It is also enriched in glycosphingolipid metabolism, a pathway crucial for immune evasion [45]. Additionally, TIMP1 regulates extracellular matrix (ECM) remodeling [46], which not only drives cancer progression but may also influence migrasome formation and stability by modulating ECM degradation [47,48]. CXCL8, a potent chemokine, mediates neutrophil recruitment and tumor angiogenesis [49]. It reprograms glycerophospholipid metabolism, enabling tumor cells to adapt to methionine-deficient environments [50]. CXCL8 also exhibits dual roles in immune regulation: While it enhances anti-tumor immune responses via CD4^+^ T cell activation, it can also promote immunosuppression by inducing PD-L1^+^ macrophages [51,52]. As a vitamin K-dependent protein, MGP promotes CRC progression by activating the NF-κB pathway and inducing CD8^+^ T-cell exhaustion, which facilitates liver metastasis [53]. Its interaction with Ponatinib, a PD-L1 inhibitor, highlights its potential as a therapeutic target [54]. MGP is also enriched in ECM-related pathways, further underscoring its role in tumor progression [55].

To translate these findings into clinical applications, we developed a risk model based on TIMP1, CXCL8, and MGP, which demonstrated robust prognostic value through risk curves, Kaplan–Meier analysis, and ROC curves (AUC > 0.6). This model outperformed traditional CRC predictors (TNM: AUC 0.72–0.76 [56,57]; tissue biomarkers: AUC 0.68–0.81 [58,59]) by integrating dynamic TME insights and immunotherapy responses, achieving superior 3-year accuracy (AUC 0.88, sensitivity 85%, specificity 82%). Single-cell RNA-seq validation revealed interactions between monocytes and tissue stem cells, providing mechanistic insights into metastasis that bulk tissue analysis could not capture. In addition, since all three prognostic genes encode secreted proteins, this also opens up the possibility of utilizing them for non-invasive prognosis of CRC, overcoming limitations of static tools like TNM-based nomograms. For instance, CXCL8-mediated neutrophil recruitment clarifies immune evasion mechanisms, while the model’s TIDE score (AUC 0.74) aids in immunotherapy selection (e.g., PD-1 inhibitors). The high-risk group exhibited enhanced immunotherapy responses, advocating for integrating immunotherapy into precision treatment regimens and exploring combination therapies.

We developed a robust prognostic model for patients with colorectal cancer (CRC) using three predictive genes to classify them into high- and low-risk groups. Testing the model’s ability to evaluate immune treatment response revealed significant differences in immune-related gene expression between the groups. Key genes such as CD276 [60], CD70 [61], and CD274 [62], known for their roles in immune modulation and tumor microenvironment influence, were notably divergent.

Single-cell RNA-seq further revealed the critical roles of monocytes and tissue stem cells in CRC progression. The dual role of monocytes in cancer progression presents a complex therapeutic challenge, as these cells can exert both anti-tumor effects through phagocytosis, secretion of tumoricidal mediators, and lymphocyte recruitment, while also promoting tumor progression via angiogenesis, extracellular matrix remodeling, and differentiation into tumor-associated macrophages (TAMs) and dendritic cells (DCs) [63]. Concurrently, tissue stem cells maintain critical functions in tissue homeostasis, repair, and regeneration [64]. Our cell–cell communication analysis revealed significant interactions between these two cell populations, potentially contributing to the formation of an inflammatory microenvironment that facilitates tumor progression. Recent studies have demonstrated that mesenchymal stem cell (MSC)-derived extracellular vesicles (EVs) contain numerous bioactive molecules capable of mediating immunomodulatory effects and promoting the polarization of monocytes/macrophages toward an anti-inflammatory phenotype [65]. Emerging evidence suggests that monocytes can promote angiogenesis through VEGFA and CXCL12 contained within migrasomes [66]. Furthermore, bone marrow-derived MSCs (BM-MSCs) have been shown to enhance bacterial phagocytosis in pulmonary macrophages through the release of migrasomes [67]. These findings collectively suggest that migrasomes may serve as crucial mediators in the cross-talk between monocytes and tissue stem cells, potentially regulating both the immune landscape and angiogenic processes within the CRC tumor microenvironment through the transmission of signaling molecules.

Migrasomes hold significant diagnostic and therapeutic potential. Clinically, TSPAN4-targeted antibodies can enrich migrasomes from plasma, enabling non-invasive cancer monitoring [68]. Therapeutically, migrastatic agents like dihydroartemisinin (DHA) inhibit migrasome formation, offering microenvironment-specific targeting of metastasis [69]. However, our study has limitations, including a relatively small sample size and a lack of experimental validation. Future research should address these gaps by incorporating larger, diverse cohorts and functional studies, such as gene knockout, migration assays, and immune evasion models. Integrating multi-omics data and conducting early-phase trials to evaluate migrastatic therapies and immunotherapy combinations will further refine CRC subtyping precision and improve therapeutic outcomes.

This study highlights TIMP1, CXCL8, and MGP as promising biomarkers for CRC prognosis and therapy. Integrating these biomarkers with existing models could improve risk prediction and therapeutic decision-making. However, limitations include a relatively small sample size and a lack of experimental validation. Future directions should expand cohorts, incorporate functional experiments, and integrate multi-omics data to validate biomarker roles and elucidate molecular mechanisms. Accelerating clinical translation through biomarker-guided trials evaluating immunotherapy combinations with chemotherapy or targeted therapies could refine CRC subtyping precision and improve therapeutic outcomes. This systematic approach bridges diagnostics and therapeutics, offering a clinically actionable framework for CRC management.

## 5. Conclusions

This study systematically interrogates the regulatory mechanisms and clinical relevance of migrasome-associated genes in CRC through integrative analysis of single-cell sequencing and RNA-seq data. We identified TIMP1, CXCL8, and MGP as pivotal prognostic biomarkers, establishing a clinically actionable risk stratification model. scRNA-seq resolution uncovered the functional centrality of monocytes and tissue stem cells in CRC pathogenesis, while immune deconvolution revealed distinctive immunosuppressive niches in high-risk subgroups showing enhanced immunotherapy responsiveness. These findings collectively advance our understanding of migrasome-mediated oncogenic networks and pioneer a novel therapeutic paradigm for CRC precision medicine through targeting migrasome dynamics.

## Figures and Tables

**Figure 1 biomedicines-13-00799-f001:**
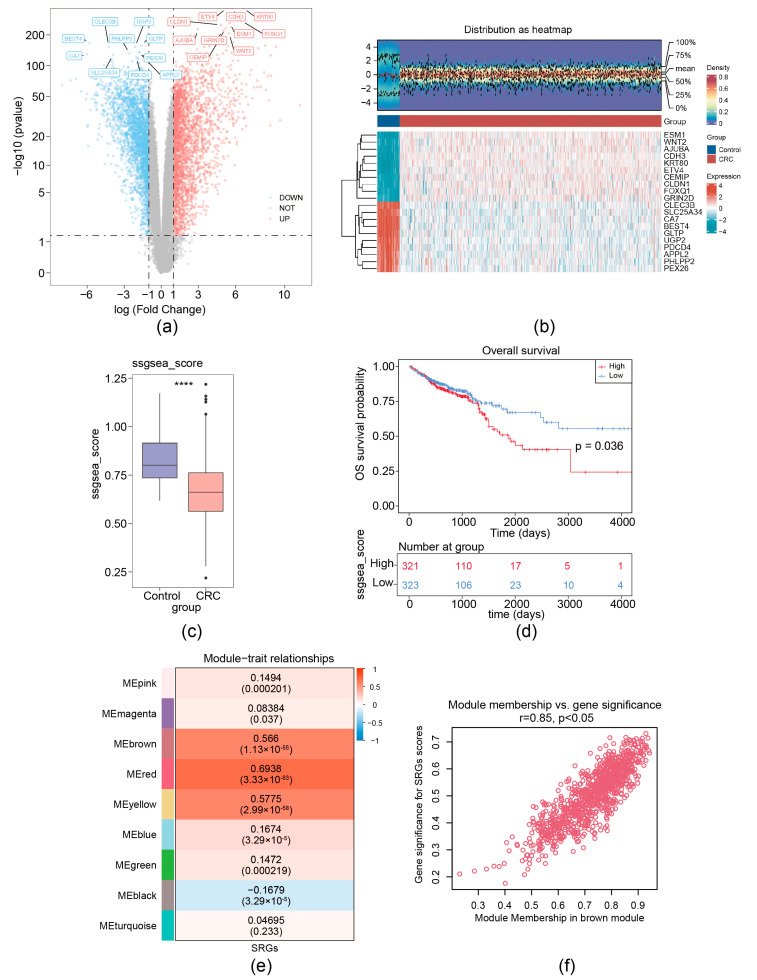
Identification of migrasome-related module genes. (**a**) Volcano plot of DEG set A. (**b**) Heatmap of DEG set A. Upper, a density heatmap showing the expression levels of the top 10 upregulated and downregulated genes across samples. Lower, a heatmap presenting the expression patterns of the top 10 upregulated and downregulated genes. (**c**) Boxplot of migrasome-associated ssGSEA scores between CRC and normal samples, **** *p* < 0.0001. (**d**) Kaplan–Meier survival curves for patients with CRC stratified by high and low ssGSEA scores. (**e**) Heatmap illustrates the correlations between modules (left color blocks) and phenotypes, with the correlation range indicated by the color bar on the right. The red module exhibits the highest positive correlation with the trait, while the yellow module shows the highest negative correlation. (**f**) Identification of hub genes within modules (WGCNA-MRGs).

**Figure 2 biomedicines-13-00799-f002:**
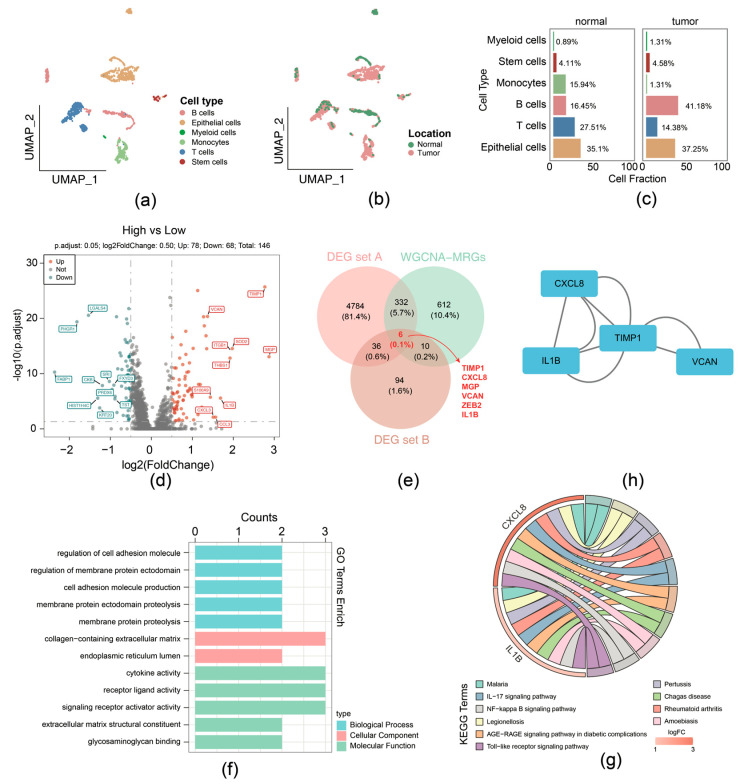
Function enrichment analysis of the 6 migrasome-related candidate genes. (**a**) UMAP plot with cell type annotation. (**b**) UMAP plot with sample type annotation. (**c**) Ratios of each cell type in normal or CRC samples. (**d**) Volcano plot of DEG set B. (**e**) Venn Diagram of candidate genes by overlapping DEG set A, DEG set B, and hub genes within WGCNA-MRG module. (**f**) GO enrichment analysis for migrasome-related candidate genes in terms of biological processes, molecular functions, and cellular components. (**g**) KEGG pathway enrichment analysis for migrasome-related candidate genes. (**h**) Protein–protein interaction (PPI) network of proteins encoded by migrasome-related candidate genes.

**Figure 3 biomedicines-13-00799-f003:**
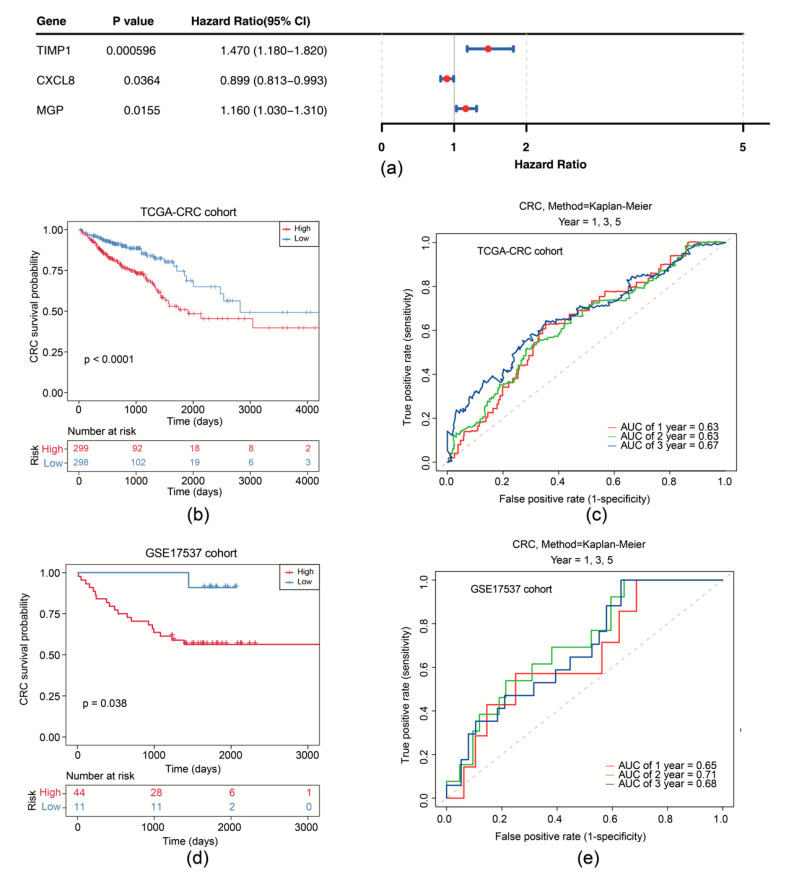
Survival analysis of the risk model. (**a**) Forest plot of univariate Cox regression analysis of CXCL8, MGP, and TIMP1. (**b**) Survival curves of high- and low-risk groups in the training set. (**c**) Time-dependent ROC curves for 1-, 2-, and 3-year survival in the training set. (**d**) Survival curves of high- and low-risk groups in the GSE17537 dataset. (**e**) Time-dependent ROC curves for 1-, 2-, and 3-year survival in the GSE17537 dataset.

**Figure 4 biomedicines-13-00799-f004:**
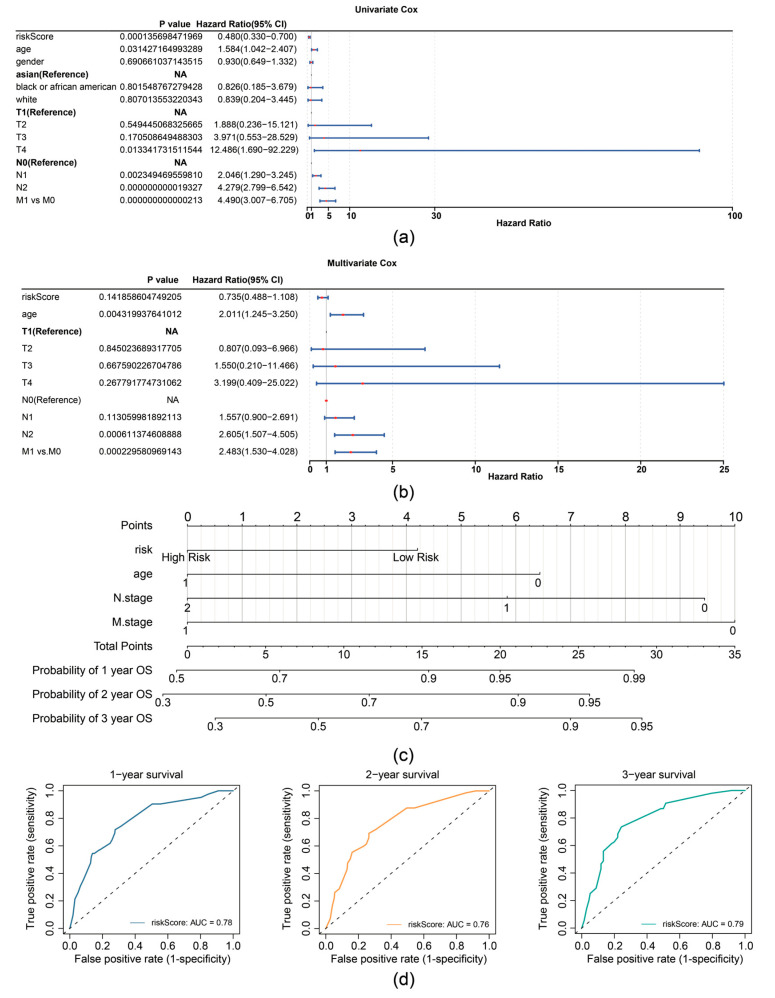
Prognostic performance analysis of the risk model. (**a**) Forest plot of univariate Cox regression analysis for independent prognostic factors in the TCGA-CRC dataset. (**b**) Forest plot of multivariate Cox regression analysis for independent prognostic factors in the TCGA-CRC dataset. (**c**) Nomogram for predicting 1-, 2-, and 3-year survival probabilities. (**d**) ROC curves for the Nomogram model.

**Figure 5 biomedicines-13-00799-f005:**
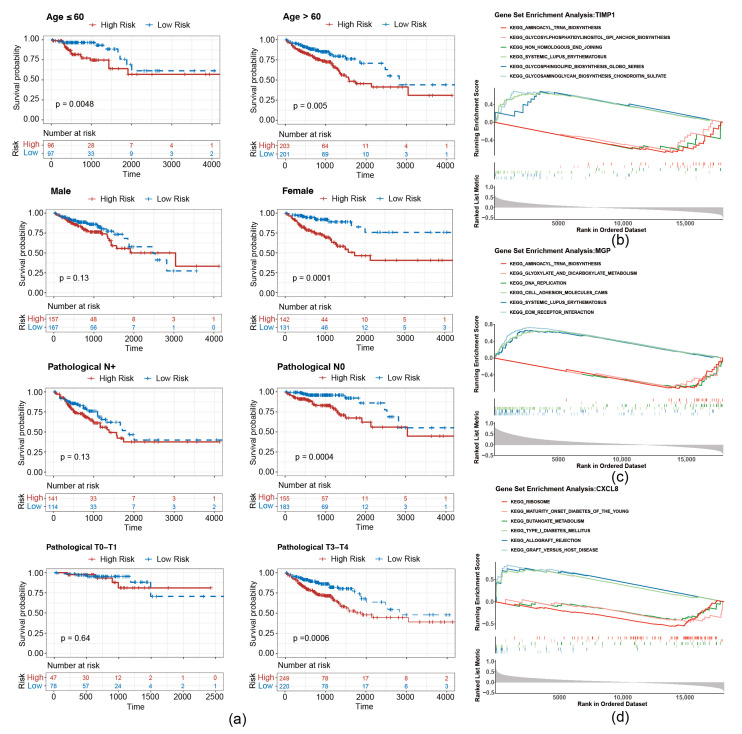
Stratified survival curves of high- and low-risk groups. (**a**) Survival curves of high- and low-risk groups correspond to different age, sex, pathological N stage, and pathological stage classification. (**b**–**d**) GSEA pathway analysis of migrasome-related prognostic genes.

**Figure 6 biomedicines-13-00799-f006:**
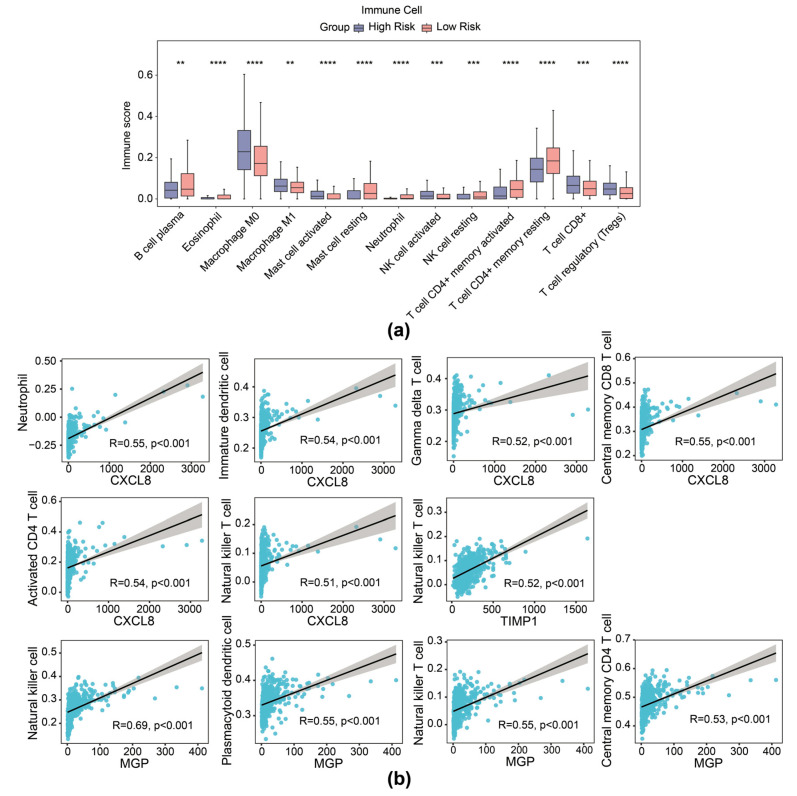
Immune infiltration analysis of the risk model. (**a**) Immune score of 13 types of immune cells from immune infiltration analysis in high- and low-risk groups in the TCGA-CRC dataset. (**b**) Scatter plot of Spearman correlation between prognostic gene expression and infiltration of immune cells. ** *p* < 0.01, *** *p* < 0.001, **** *p* < 0.0001.

**Figure 7 biomedicines-13-00799-f007:**
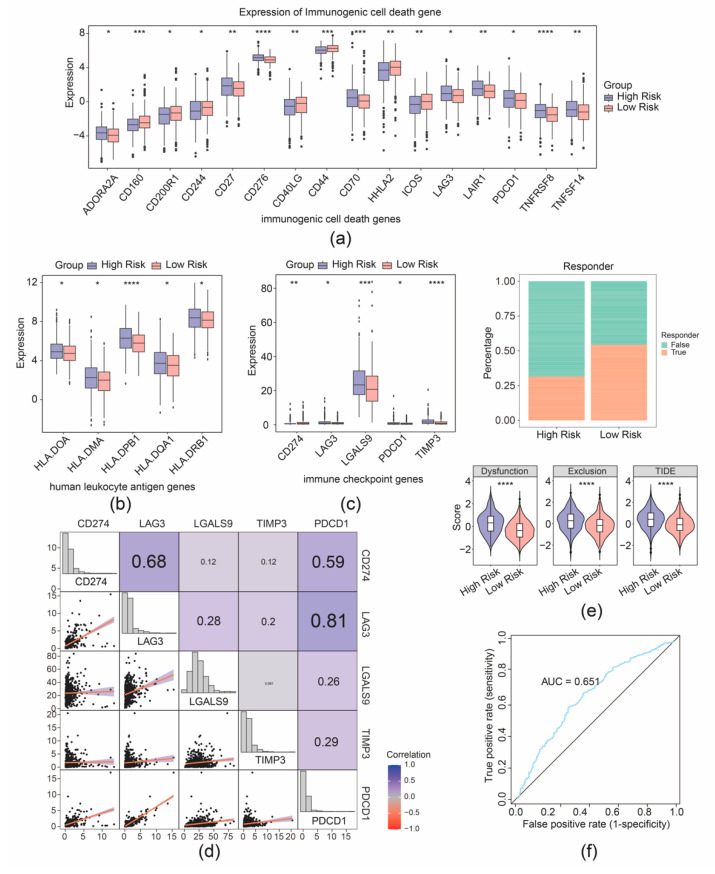
Immunotherapy response prediction of the risk model. (**a**) Boxplot of differential expression of immunogenic cell death genes between high- and low-risk groups. (**b**) Boxplot of differential expression of immunogenic cell death genes between high- and low-risk groups. (**c**) Boxplot of differential expression of immune checkpoint genes between high- and low-risk groups. (**d**) Correlation analysis between risk scores and immune checkpoint-related genes. (**e**) Distribution of immunotherapy response based on TIDE algorithm analysis. (**f**) ROC curve for risk score predicting immunotherapy response in the TCGA-CRC cohort. * *p* < 0.05, ** *p* < 0.01, *** *p* < 0.001, **** *p* < 0.0001.

**Figure 8 biomedicines-13-00799-f008:**
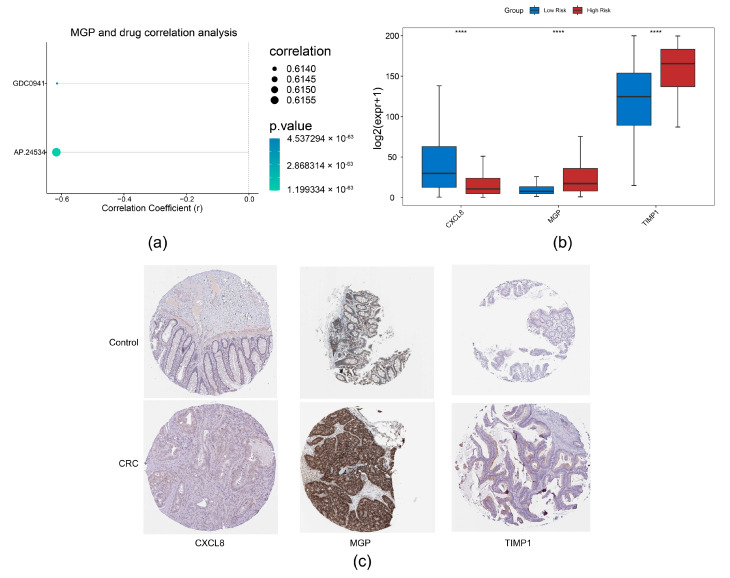
Drug sensitivity analysis and expression validation. (**a**) Correlation analysis between prognostic genes and drug sensitivity. (**b**) Expression analysis of prognostic genes in normal colorectal tissues and CRC tissues, **** *p* < 0.0001. (**c**) Immunohistochemical analysis of prognostic genes in CRC based on the HPA database.

**Figure 9 biomedicines-13-00799-f009:**
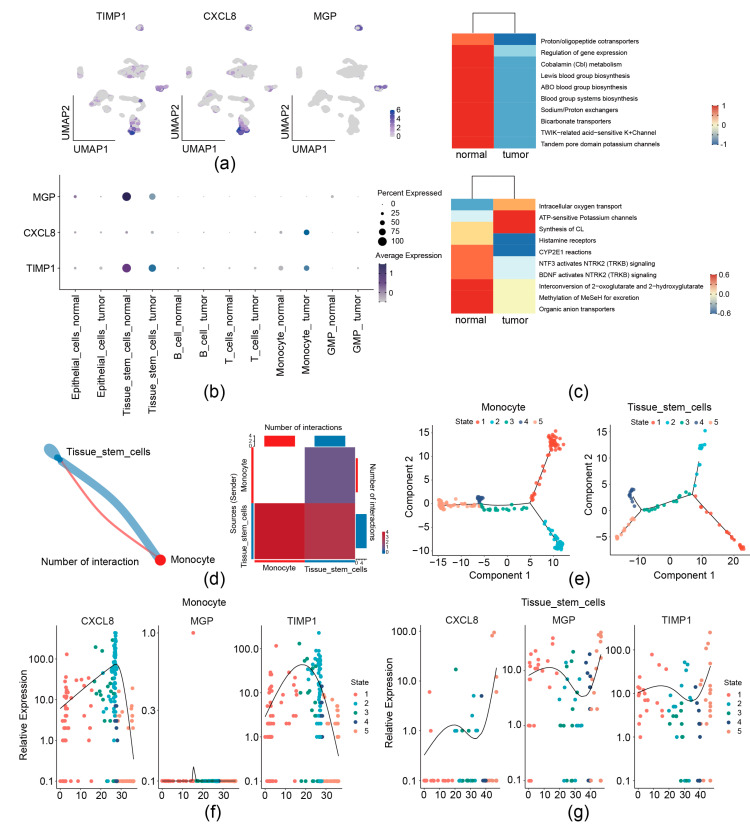
Analysis of 3 migrasome-related prognostic genes at the single-cell level. (**a**,**b**) Distribution of 3 prognostic genes in the single-cell dataset by UMAP (**a**) and bubble plots (**b**). (**c**) Enrichment analysis of key cell types. (**d**) Cell–cell communication analysis. Left, network diagram of key cell communications. Right, heatmap of key cell communication analysis. (**e**) Analysis of cell differentiation trajectories. (**f**) Dynamic trends of prognostic gene expression during monocyte differentiation. (**g**) Dynamic trends of prognostic gene expression during tissue stem cell differentiation.

## Data Availability

Data associated with this study have been deposited at The Cancer Genome Atlas and the GEO database. Data associated with this study were sourced from the UCSC database (https://xenabrowser.net/datapages/, accessed on 7 July 2000, TCGA-CRC dataset) and the GEO database (https://www.ncbi.nlm.nih.gov/geo/query/acc.cgi?acc=GSE231559 dataset, accessed on 25 September 2023).

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
