# Peer review of "Migrasome-Related Genes as Potential Prognosis and Immunotherapy Response Predictors for Colorectal Cancer"

_biomedicines, 2025, doi:10.3390/biomedicines13040799_

Round 1

Reviewer 1 Report

Comments and Suggestions for Authors

This work investigates migrasomes in CRC, an underexplored area, providing new insights on their role in tumor progression and immune modulation. The incorporation of migrasome-associated genes into a prognostic model provides a clinical context, which likely will help to inform on risk and treatment options. The research is organized effectively and follows sound methodology, yet it needs greater clarity, more robust mechanistic descriptions, and clearer language to boost readability.

- please improve the conclusion, it is way too short for a research paper and state the future direction over there.

-The effect of migrasomes on the tumor microenvironment requires a more precise mechanistic clarification with additional specific references. Lack of validation experimental. The study is entirely dependent on bioinformatics and computational analysis, supported by no experimental validation (e.g., qPCR, Western blot, functional assays).

- The proposed risk model must work in independent clinical cohorts or patient-derived samples to hold promise for predictive accuracy.

- signaling pathways or their functional impact on cell migration, invasion, or immune evasion) is needed.

- Also, the model does not explore the connection of monocyte, tissue stem cell, and migrasome interactions in CRC fully, leaving voids in understanding their contribution to tumor progression.

  • Immune checkpoint markers (CD276, CD70, CD274) are brought up, but the analysis of any correlation of their expression to treatment response or survival was shallow.

-The study posits that migrasomes could serve as promising biomarkers or therapeutic targets; without consultation on how such discoveries will be applied for clinical obvious purposes (companion diagnostics, therapeutic interventions).

-A more vigorous comparison of the model against published prognostic models would greatly enhance the trial-governed directions of this study.

- overall, lack of clarity in the discussion

- Compare and contrast maximal functions for CRC with different cancers, pointing out the uniqueness or novelty of CRC in this regard.

- How does the model obtained in this study compare with already existing CRC prognostic models? If any advantage is there (for example, better prediction accuracy, or integration with immune response markers), make it more explicit.

Reviewer 2 Report

Comments and Suggestions for Authors

This paper investigates the role of microsome-related genes (MRGs) in colorectal cancer (CRC) prognosis and immunotherapy response prediction. The authors employ bioinformatics approaches, including differential gene expression analysis, weighted gene co-expression network analysis (WGCNA), single-cell RNA sequencing (scRNA-seq), and functional enrichment analysis, to identify key MRGs associated with CRC. They establish a prognostic risk model based on three genes (TIMP1, CXCL8, and MGP) and validate its ability to stratify CRC patients into high-risk groups. The study also explores immune infiltration patterns and immunotherapy responses linked to these genes, proposing that microsomes contribute to CRC progression and tumor immune microenvironment regulation. This research highlights the potential of MRGs as biomarkers for CRC prognosis and therapeutic targets. However, there are still some issues that need to be addressed.

  1. The single-cell RNA-seq analysis identifies monocytes and tissue stem cells as key cell types, but the biological implications of their interaction with migrasomes in CRC are not sufficiently discussed.
  2. The paper does not explore the biological mechanisms of TIMP1, CXCL8, and MGP on CRC cell proliferation, migration, or immune evasion. It’s recommended to conduct pathway enrichment and functional assays (e.g., migration and apoptosis assays) to investigate how these genes influence CRC tumorigenesis and the tumor immune microenvironment.

  3. Although migrasomes are central to the study, their formation, release, and role in CRC are not explored in depth. A discussion on how CRC-specific factors influence migrasome dynamics would be valuable.
  4. It’s recommended to perform a comparison between the proposed model and other widely used models to highlight its unique advantages or potential limitations in CRC progression.

Reviewer 3 Report

Comments and Suggestions for Authors

The article entitled “Migrasome-related genes as potential prognosis and immunotherapy response predictors for colorectal cancer” debate a particularly important and actual topic for 21st century society. Unfortunately, even though medicine has advanced a lot in the last decade, there are still pathologies and problems without answers or with incomplete detection and treatment methods. Thus this study may represent an important ace in the pathology of colorectal cancer. Basically the article the study investigates migrasome-related genes as potential biomarkers for colorectal cancer (CRC) prognosis and immunotherapy response. Using bioinformatics analysis of TCGA and GEO datasets, authors identified TIMP1, CXCL8, and MGP as key prognostic genes. A risk model based on these genes effectively stratified patients into high- and low-risk groups, with high-risk patients showing lower survival rates but better response to immunotherapy. Single-cell RNA sequencing revealed that these genes are highly expressed in monocytes and tissue stem cells, playing a role in tumor progression and immune regulation.

Even if the article presents interesting and important information, I have some concerns that I would like to express below:

Query 1: The study relies heavily on bioinformatics and computational analyses (TCGA, GEO, scRNA-seq). So the results don’t have in vivo or in vitro validation. I want to know if you think that other experiments are needed and if you think that other determinations can influence the presented results.

Query 2: Another concern is related to the number of samples examined. The scRNA-seq dataset includes only six CRC and three control samples, which may not fully represent CRC heterogeneity. How can you explain this aspect?

Query 3: Another detail is about the citation style in the text. The citation style don’t follow the journal guidelines.

Query 4: Please totally revise the reference list. Again, the reference list does not comply with the journal guidelines.

Query 5: All the results of research must have practical implications for society. So, I’m wondering, how can your findings be translated into clinical practice for CRC prognosis and treatment decision-making?

Query 6: Please add in the article the most important limitations of the present study.

In the end I want to congratulate the authors for their work and I want to wish them good luck in the future!

Reviewer 4 Report

Comments and Suggestions for Authors

Overall evaluation

The study comprehensively uses multiple bioinformatics methods for analysis from multiple levels, and the methods are relatively comprehensive. However, there are some issues with the application of certain methods. During the data processing, the basis for selecting some key parameters is insufficient. When determining the soft threshold in the Weighted Gene Co-expression Network Analysis (WGCNA), only the screening results are given, but the impact of choosing this threshold on subsequent results and the comparison with other possible thresholds are not elaborated in detail, making the reliability of the method doubtful.

The research results are relatively rich, and the constructed risk model and related analyses have certain reference value. However, there are some defects in the presentation and interpretation of the results. In the part of immunotherapy response prediction, although differences in immune-related indicators between the high-risk group and the low-risk group have been found and a better immunotherapy response rate in the high-risk group has been observed, there is a lack of in-depth exploration of how these differences can be directly translated into guidance for clinical treatment decisions. At the same time, when presenting the results of univariate and multivariate Cox analyses, some data in the forest plot are not clearly marked, affecting the accurate understanding of the results.

The study aims to explore the potential of migrasome-related genes as predictors of prognosis and immunotherapy response, which has certain clinical significance. However, currently, the research only stays at the bioinformatics analysis level, lacking the verification of clinical samples and the support of clinical trials. There is still a large gap from practical clinical application, and it cannot provide reliable diagnosis and treatment basis for clinicians.

Specific problems and suggestions for improvement

  1. The article mentions that 7 migrasome-related genes (MRGs) were identified from the literature, but the specific basis and screening criteria for these genes were not described in detail. It is recommended to supplement the detailed description of the screening process of these 7 genes, including the scope of literature, key screening indicators, etc., so that readers can better understand the starting point of the research and the rationality of gene selection.
  2. When determining the soft threshold in the WGCNA analysis, only the result of "\(\beta = 26\) (\(R^{2}\) of the scale - free network = 0.85)" is given, and the reasons for choosing this threshold and the comparison with other possible thresholds are not elaborated in detail. It is recommended to supplement the changes in the network topology under different thresholds, such as the impact of different thresholds on module division, gene connectivity, etc., to fully illustrate the rationality and advantages of choosing this threshold.
  3. In the functional enrichment analysis, the interpretation of the results of KEGG and GO analyses is relatively simple. Only some related biological processes and signaling pathways are listed, and the internal connections between these results and the occurrence, development of colorectal cancer, and the function of migrasomes are not explored in depth. It is recommended to further explore the enrichment results, and combined with existing research, elaborate on the mechanism of action of these biological processes and signaling pathways in colorectal cancer in relation to migrasome-related genes. For example, how these pathways affect the progression of colorectal cancer by influencing the tumor microenvironment.
  4. When constructing the risk model, only the TCGA - CRC cohort was used for model development and preliminary verification, and the sample source is relatively single. It is recommended to add other independent colorectal cancer cohorts for external verification, such as samples from different ethnic groups and regions, to improve the universality and reliability of the risk model.
  5. When drawing the nomogram, the basis for selecting the clinical covariates (age, gender, N, T, and M stages) included in the nomogram was not described in detail. It is recommended to supplement the statistical analysis and clinical significance of the selection of these covariates, explain why these factors are selected to be included in the nomogram, and the rationality of their combination with the risk model.
  6. In the part of immunotherapy response prediction, although differences in immune cell infiltration and immune-related gene expression between the high-risk group and the low-risk group have been found, there is a lack of in-depth exploration of how these differences can be translated into guidance for clinical immunotherapy decisions. It is recommended to add specific suggestions on the selection of immunotherapy regimens and efficacy evaluation for patients in different risk groups. For example, which immunotherapy drugs may be more suitable for high-risk and low-risk group patients, and how to adjust the dose and course of immunotherapy according to the risk score.
  7. The results of univariate and multivariate Cox analyses are presented in the forest plot, but some data are not clearly marked. For example, the confidence intervals of the hazard ratio (HR) are marked vaguely, and some P-values do not clearly show the specific values. It is recommended to redraw the forest plot to ensure clear and accurate data marking, so that readers can more intuitively understand the analysis results.
  8. The research is only based on bioinformatics analysis and lacks experimental verification of clinical samples, such as immunohistochemistry, qRT - PCR and other methods to verify the screened migrasome-related genes. It is recommended to supplement the experimental data of clinical samples to further verify the expression differences of migrasome-related genes in colorectal cancer tissues and normal tissues, as well as their correlations with patient prognosis and immunotherapy response, thereby improving the credibility and clinical application value of the research results. In addition, please perform cell and animal experiments and combine with clinical samples for further experimental validation.

Comments on the Quality of English Language

The English could be improved to more clearly express the research. 

Round 2

Reviewer 1 Report

Comments and Suggestions for Authors

Thank you for answering all my questions. Good luck 

Reviewer 3 Report

Comments and Suggestions for Authors

Dear authors,

Thank you for considering all my suggestions.

I have one last suggestion. Because of the limited number of samples I strongly recommend you to submit the current article in form of a "Communication".

Thank you!

Reviewer 4 Report

Comments and Suggestions for Authors

I have no further questions.